cellular biology/evolution

cancer evolution, heritability, phenotypic inheritance, experimental evolution

**Author for correspondence:**
Louise J. Johnson
e-mail: l.j.johnson@reading.ac.uk

# Cancer cell lines show high heritability for motility but not generation time

Anastasia V. Wass[1], George Butler[1], Tiffany B. Taylor[1,2], Philip R. Dash[1] and Louise J. Johnson[1]

[1]School of Biological Sciences, University of Reading, Whiteknights, Reading, Berkshire RG6 6AH, UK
[2]The Milner Centre for Evolution and Department of Biology and Biochemistry, University of Bath, Claverton Down, Bath, Somerset BA2 7AY, UK

 GB, 0000-0002-6207-6225; TBT, 0000-0002-5274-7806;
PRD, 0000-0002-6029-4560; LJJ, 0000-0002-0006-1511

Tumour evolution depends on heritable differences between cells in traits affecting cell survival or replication. It is well established that cancer cells are genetically and phenotypically heterogeneous; however, the extent to which this phenotypic variation is heritable is far less well explored. Here, we estimate the broad-sense heritability ($H^2$) of two cell traits related to cancer hallmarks—cell motility and generation time—within populations of four cancer cell lines *in vitro* and find that motility is strongly heritable. This heritability is stable across multiple cell generations, with heritability values at the high end of those measured for a range of traits in natural populations of animals or plants. These findings confirm a central assumption of cancer evolution, provide a first quantification of the evolvability of key traits in cancer cells and indicate that there is ample raw material for experimental evolution in cancer cell lines. Generation time, a trait directly affecting cell fitness, shows substantially lower values of heritability than cell speed, consistent with its having been under directional selection removing heritable variation.

## 1. Introduction

Evolutionary processes are acknowledged to play a significant role in the initiation and progression of cancer, as well as in the acquisition of traits such as chemotherapy resistance [1–3]. Cancer evolution as a field rests on the reasonable, but rarely directly tested, assumption that a variety of cellular traits linked to cancer development and progression are heritable at the level of the cell population: that is, there is phenotypic variation between cancer cells, and that at least some of that variation is due to factors passed on from mother cell to daughter cell, rather than being caused by environmental factors such as nutrient availability. For example, explaining the evolution of metastatic

behaviour in terms of dispersal or foraging ecology requires that cells should vary heritably in their dispersal behaviours [4], and models of intra-tumour competition [5], by definition, assume a heterogeneous population with heritable differences between cells. The heterogeneity of cancer cells, whether in tumours or laboratory cell lines, is well established—for instance, a recent multi-omics study found high levels of both genetic and phenotypic heterogeneity between different populations of HeLa cells [6]. Using microfluidic devices, it has also been possible to induce and measure migration of single cells [7] (Yan & Irimia 2014) and to isolate subpopulations for analysis, between which stable phenotypic differences are seen [8,9]. However, heritability—which is defined in quantitative genetics as the proportion of trait variance in a population that is due to genetic variation, and which determines the response to selection [10]—has never been directly measured for any trait in cancer cell populations.

Drastic reductions in the cost of sequencing technology over recent years have provided the cancer research community with an abundance of genome sequence data, including at a single-cell level [11]. Sequence analysis techniques adapted from population and evolutionary genetics can detect and measure evolutionary processes in tumours [12], and signatures of natural selection can be detected by various methods including comparing synonymous and non-synonymous substitution rates [13], or estimating the relative contributions of neutral and adaptive evolution using distributions of allele frequencies within tumours [14]. However, while such analyses can show natural selection has occurred and sometimes identify target genes, they are not always informative as to which phenotypic traits are the targets of selection. Researchers increasingly recognize the need to complement sequence data with a clear and quantitative understanding of cell phenotypes, and—where possible—to link genotype to phenotype [15]. Theoretical models and simulations of cancer evolution, both population level and agent based, will undoubtedly guide this programme of phenotypic research [16], but these approaches could be even more valuable if informed by cell-level observational data.

Based on observations of cancer cells over multiple cell generations, we here present cell-level phenotypic data that allows us to estimate broad-sense heritability ($H^2$) of two traits within four clonally reproducing cancer cell lines. Generation time—the time elapsed between cell divisions—is closely linked to cell fitness. Cell motility is a key step toward metastasis, but confers an uncertain [17,18], and likely context-dependent [19], selective advantage.

Both generation time and cell motility are observable and quantifiable using time-lapse video microscopy (figure 1*a*). Cell lineages must be observed over several generations to give reliable estimates of heritability, because cytoplasmic factors cause transient similarity between sister cells for several hours after cell division [20], which will inflate estimates. There is also the possibility that soon after cell division, over short spatial scales, sister cells could affect one another's behaviour via secreted signals, further increasing the similarity between sister cell pairs. We therefore estimate $H^2$ based on several different cell–cell relationships. By tracking and comparing second-generation clonal descendants of the same progenitor cell ('cousin cells'; figure 1*b*), which have never directly shared cytoplasm, we can provide the estimates of $H^2$ that represent stably inherited differences between clonal lineages.

# 2. Material and methods

## 2.1. Cell lines

All cell lines used were laboratory adapted. Three are adenocarcinoma in origin: MCF7 (ATCC® HTB-22™) [21], MDA-MB-231 (ATCC® HTB-26™) [22] and HeLa (ATCC® CCL-2™) [23]. HT1080 (ATCC® CCL-121™) [24] are fibrosarcoma derived. Cell lines were grown as a monolayer in 5% $CO_2$ at 37°C in minimum essential media containing 10% foetal bovine serum, 1 mM sodium pyruvate and 2 mM glutamine. MCF7, HT1080 and HeLa cell media also contained 1% non-essential amino acid solution. Passage numbers were 15–18 for MCF7, 41–44 for MDA-MB-231 and 5–8 for both HeLa and HT1080 cells. Typical laboratory cell culture maintains cell lines at population sizes of $10^5$ to $10^6$.

## 2.2. Time-lapse microscopy and lineage tracking

In total, 9025 cells were tracked, giving a dataset of 471 573 cell positions at known time points. For each of 59 time-lapse videos taken, a haemocytometer was used to plate 5000 cells per well onto a 24-well plate. Six wells per cell line were distributed around the plate. Five points within each well were chosen at random and images were taken every 20 min over 72 h with a Nikon TiE Time Lapse

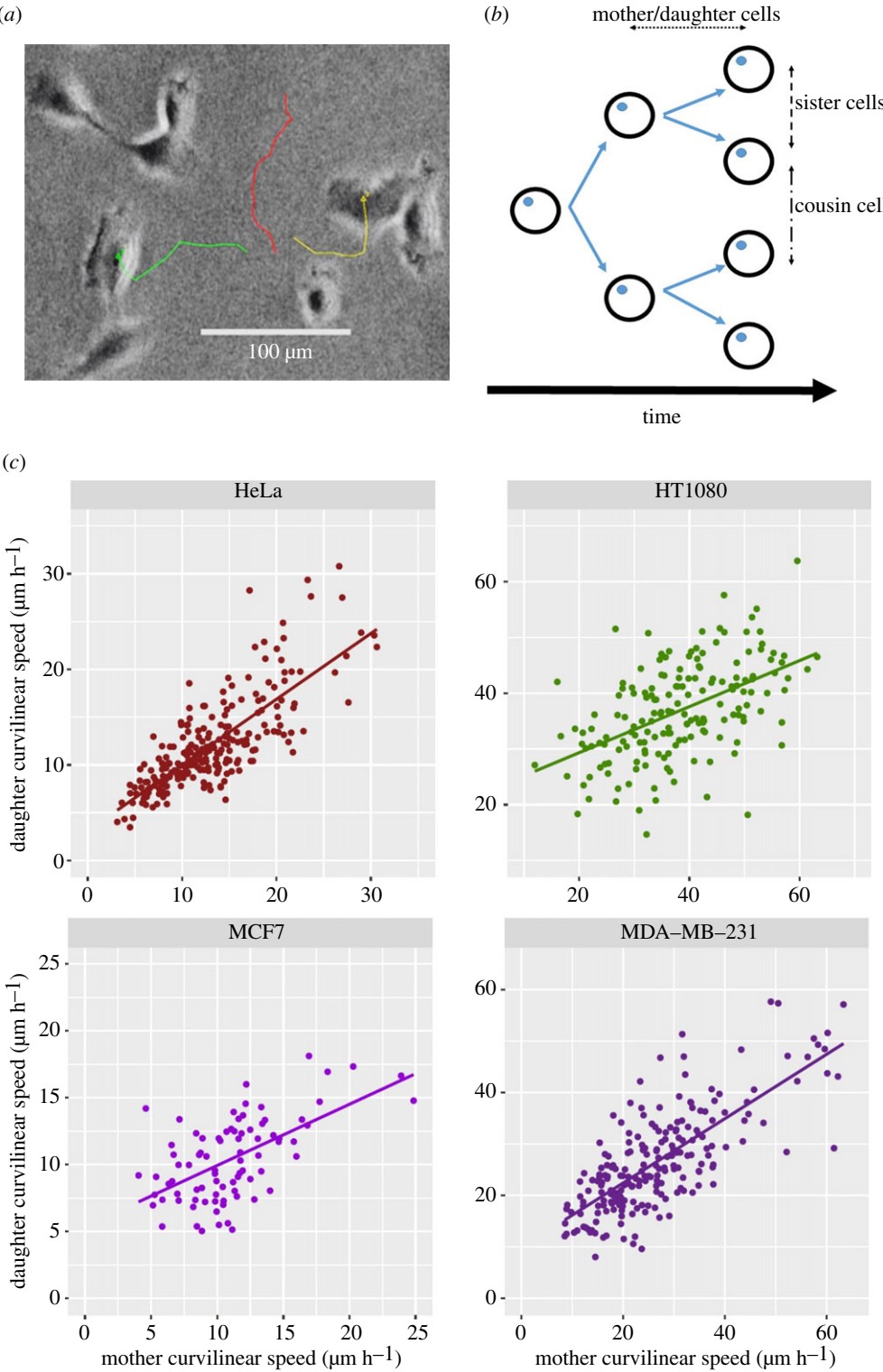

**Figure 1.** (*a*) Example image from a time-lapse video, showing one HT1080 cell lineage as tracked for 11 h 40 min during the 72 h observation period. Here, a mother cell moved from the upper middle part of the frame to the lower middle (red line) and divided into two daughter cells, which have since moved outwards to the left and right (green and yellow lines). See electronic supplementary material, Files for video. (*b*) Schematic diagram of a cell family over three cell divisions showing the three cell–cell relationships used to calculate broad-sense heritability. (*c*) Example of parent–offspring regression for motility, shown here for the mother cell/daughter cell relationship, in all four cell lines tested. Each point represents a mother cell and the mean speed of her daughters, and the slope of the regression line is the estimate of broad-sense heritability. Note that axes differ between graphs due to differences in speed between cell lines. As the cells reproduce clonally, the same method is applicable to other cell–cell relationships. $H^2$ values for all cell–cell relationships are shown in table 1.

system. NIS software was used to convert images into a video file for each point. ImageJ and MtrackJ [25,26] were used to analyse videos, recording cell positions at each timepoint and cell division events. Our final dataset consisted of cell families: groups of related cells for which we had whole-lifespan data on cell motility and generation time over three cell generations (figure 1b). 2048 individual cells (22.7% of cells originally tracked) could usefully be included in our analysis, in that they were members of cell families in which it was possible to compare at least one pair of 'cousin cells'. These comprised 52 families for the cell line MCF7, 141 families for MDA-MB-231, 134 families for HeLa and 110 families for HT1080.

## 2.3. Statistical analysis

Statistical analysis was performed in R [27]. The generation time for a given cell is the time taken for a full cell cycle from cytokinesis to cytokinesis. Cell motility was calculated as the curvilinear speed of a cell over its entire individual lifetime: the total Euclidean distance of the path travelled from cell division to cell division, in microns, divided by its generation time in hours. All cells included in our analysis were observed for a complete cell cycle; the cells originally plated were therefore excluded, as were those that moved off-screen, died or had not divided by the end of the tracking period. For this reason, some cell families contained fewer than six cells. We could detect no significant differences in cell speed between families containing different numbers of cells, and there was also no significant difference in cell speed between individual cells that moved off-screen compared to those that were tracked until division (t-tests, $p > 0.05$ in every cell line for each test). While it is still possible that we have failed to track some proportion of faster cells that were more likely to move off-screen over the tracking period, these results suggest that such an effect is unlikely to be large. There was also no significant difference between wells (Kruskal–Wallis test; $p > 0.05$), suggesting that spatial effects—which could exacerbate similarities between related cells and inflate estimates of heritability—are minimal. We then estimated broad-sense heritability as the slope parameter of a simple least squares regression of trait values between clonal cells and their relatives, for all three cell–cell relationships (figure 1c); that is, our estimate of $H^2$ is $\mathrm{Cov}(x, y)/\mathrm{Var}(x)$ where our $x$ and $y$ values are the speeds of pairs of related cells. Where a cell had multiple tracked daughters or 'cousins,' their mean value was used. This is a modification of standard parent–offspring regression techniques [28] for determining heritability in clonally reproducing cell families and provides a straightforward way to compare heritability values calculated from different cell–cell relationships; the structure of our dataset, with some wells containing very few families, made this a more appropriate method of analysis than a linear mixed model.

# 3. Results and discussion

## 3.1. Cell motility

For all four cell lines, and for all cell–cell relationships considered, the broad-sense heritability ($H^2$) of motility is highly significant (figure 1c; table 1). Values of $H^2$ ranged from 0.45 to 0.84, which is high compared to that of a range of traits in natural [29,30] and agricultural [31] populations. Although no one statistic adequately describes evolvability, this result does imply that cancer cell populations contain substantial variation for motility on which natural selection could act. Mutations, stable epimutations, or both may contribute to this heritable variation [32], and potentially to evolutionary change [33].

## 3.2. Generation time

Generation time shows a very different pattern of heritability (table 1). For all lines and all cell relationships, $H^2$ values are lower than those for motility. There is no consistent relationship between the two traits: in two cell lines, faster dividing cells were slower moving (for HeLa, Spearman's $\rho = -0.068$; $p < 0.001$; for MCF7, $\rho = -0.178$; $p < 0.005$); in HT1080, faster dividing cells were faster moving ($\rho = 0.072$; $p < 0.005$) and no significant correlation was seen in MDA-MB-231 ($\rho = -0.002$; $p > 0.1$).

Apart from in the HeLa cell line, only sister–sister heritability is consistently significant for generation time, consistent with variation in this trait being attributable to transient cytoplasmic or nutritional differences, or perhaps secreted signals—the MDA-MB-231 cell line shows only marginally significant

**Table 1.** Mean trait value (± standard deviation) and broad-sense heritability of motility and generation time, in four cell lines, for three relationships between cells in tracked clonal families. $R^2$ is also given for each regression.

| cell line | no. of families | mean speed (μm hour$^{-1}$) | mean generation time (hours) | heritability of cell motility $H^2$ ($R^2$) | | | heritability of generation time $H^2$ ($R^2$) | | |
|---|---|---|---|---|---|---|---|---|---|
| | | | | sister:sister | mother:daughter | cousin:cousin | sister:sister | mother:daughter | cousin:cousin |
| MCF7 | 52 | 10.76 ± 3.74 | 24.78 ± 5.88 | 0.5*** (0.28) | 0.46*** (0.33) | 0.58** (0.27) | 0.49*** (0.35) | 0.15 (0.01) | 0.27 (0.03) |
| MDA-MB-231 | 141 | 26.82 ± 11.37 | 22.46 ± 6.62 | 0.82*** (0.65) | 0.59*** (0.55) | 0.56*** (0.41) | 0.47*** (0.23) | 0.10 (0.00) | 0.27* (0.06) |
| HT1080 | 110 | 37.55 ± 10.03 | 14.22 ± 4.15 | 0.62*** (0.44) | 0.45*** (0.3) | 0.56*** (0.3) | 0.45*** (0.11) | 0.15* (0.02) | 0.26 (0.02) |
| HeLa | 134 | 12.48 ± 5.23 | 24.43 ± 4.46 | 0.82*** (0.6) | 0.69*** (0.61) | 0.85*** (0.60) | 0.56*** (0.27) | 0.4*** (0.08) | 0.62*** (0.36) |

$*p < 0.05$, $**p < 0.01$, $***p < 0.001$.

$H^2$ for generation time between cousins. As the low $R^2$ values indicate, the regression model explained the data poorly, suggesting there may be other factors not included in our model which would explain more of the variation.

Differences in heritability between traits could have a variety of causes: traits are likely to vary in sensitivity to environmental factors, and contributing genes may show different levels of standing variation. Generally, however, lower values of heritability might be expected for traits such as generation time which are highly correlated with fitness, as genetic variation is likely to be removed by natural selection [34]. Conversely, selection can maintain phenotypic variation as well as remove it, and one might expect to see high heritability of traits under fluctuating or frequency-dependent selection—which, under some models of evolution, would include motility traits.

The cell lines MCF7 and HeLa are both highly genetically unstable, showing extensive genome rearrangement [35], high levels of chromosome number variability [36] and many instances of regional copy number increase [37]. Such cell lines might be expected to show higher $H^2$ values, and greater evolvability, due to a greater level of standing genetic variation. This was not borne out for MCF7 in our data, although HeLa did show high $H^2$ values compared to the other cell lines tested.

## 3.3. Conclusions and future directions

Our results confirm an important but previously untested assumption of cancer evolution [38,39] and are encouraging for the prospect of experimental cancer evolution *in vitro*, an emerging field that is beginning to provide new insights into cancer biology and evolution [40–42] in the same way as microbial experimental evolution has advanced our understanding of adaptation more generally [43].

Similar methods could be used to estimate evolvability parameters in a range of traits, such as cell adhesion or extracellular matrix degradation, and a range of cell populations. It would be interesting to compare a range of cancer types with differing tendencies to metastasis—perhaps including transmissible cancers—and cancers from different patients, including recently isolated lines and biopsy samples. Primary non-cancerous cells should also be tested: currently, beyond a few measures of proliferation rate and marker gene expression in Chinese hamster ovaries [44], very sparse data are available on the extent of heritable phenotypic variation between clonal somatic cells of multicellular animals, despite this question being highly relevant to the evolution of multicellularity [45]. Yan *et al.* [7] measured the speed of individual cancer cells (including MCF-7 cells) in microfluidic channels and reported no correlation between the speeds of sister cells; the discrepancy between their results and our own may be because our statistical analysis was more sensitive, or due to differences in experimental method, such as the use of collagen coating, or the fact that in their experiment, cells were responding to self-generated ECF gradients in fine capillaries, whereas in ours, the liquid medium that covered the cells was able to mix more freely and might not have sustained sharp gradients.

Good data on trait heritability are available for a few model species of unicellular eukaryotes. Broad-sense heritability estimates obtained for 46 growth traits in *Saccharomyces cerevisiae* [46] had a median value of 0.77, and in *Neurospora crassa*, the heritability of six clock traits [47] ranged from 0.42 to 0.87. However, both of these experiments started with crosses between geographically disparate strains to maximize genetic variation. We obtain similar heritability estimates within individual clonal cell lines, suggesting that cancer cells undergo comparatively rapid phenotypic diversification.

Gene expression data from cells with varying trait values within the same cell line, or comparisons between selection and control lines after experimental evolution, could reveal possible targets of selection. Little is currently known about the molecular basis of phenotypic variation within tumours, although molecular genetic variation is known to be substantial [48,49].

Because heritability is affected by the levels of environmental variation [50], values of heritability *in vivo* are likely to differ from *in vitro* estimates. Further quantitative data on cancer cell populations either in culture conditions simulating particular conditions of interest in tumour microenvironments, or within real tumours, are needed to enable the evolutionary dynamics of cancers to be understood and translated into meaningful *in vivo* predictions for personalized medicine and to reveal new targets and therapies for future clinical interventions.

Data accessibility. All data generated and analysed in this study are available at http://dx.doi.org/10.17864/1947.167

Authors' contribution. A.V.W., P.R.D., T.B.T. and L.J.J. conceived the project. A.V.W. conducted laboratory work and collected data. G.B. and A.V.W. conducted statistical analyses. All authors contributed to writing the manuscript. P.R.D. and L.J.J. co-supervised the project and are joint senior authors. All authors gave final approval for publication and agree to be held accountable for the work performed therein.

Competing interests. No competing interests to declare.

Funding. This work was supported by the University of Reading, the Leverhulme Trust (reference: F/00 239/AL) and an anonymous charitable donation.

Acknowledgements. We thank Michael Brockhurst, Jonathan Gibbins, Robert Jackson, Michael Fry, Mark Pagel and Karen Poulter for comments. G.B. thanks tutors and participants at the 2018 Evolutionary Biology and Ecology of Cancer Summer School for helpful discussion. The authors have no conflicts of interest to declare.

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
