## [Reviewer comments · Royal Society Open Science]

Review History

RSOS-191645.R0 (Original submission)

Review form: Reviewer 1

Is the manuscript scientifically sound in its present form?

Yes

Are the interpretations and conclusions justified by the results?

No

Is the language acceptable?

Yes

Do you have any ethical concerns with this paper?

No

Have you any concerns about statistical analyses in this paper?

No

Recommendation?

Major revision is needed (please make suggestions in comments)

Comments to the Author(s)

The manuscript entitled "Cancer cell lines show high heritability for motility but not generation time" by Wass et al. tracked cell migration and proliferation of hundreds of cell family from multiple cells to test whether motility and proliferation rate are inheritable. The authors did a lot of works and provided interesting data. However, there are limitations in this study. The authors didn't compare what they found with relevant previous works. In addition, there are concerns regarding data processing/selection and alternative interpretation of the collected data. It is recommended that this paper can be revised before being accepted. The detailed comments are listed below:

1. There were some previous works regarding whether the cell motility can be inherited in the literature. The work from Yan et al is a short-term study more similar to the presented work. They saw low correlation between two daughter cells. The works from Chen et al are long-term heritability studies. They showed cellular motility and altered gene expression can be inherited. It is recommended that the authors should explicitly discuss what's new in the present work and also why the authors got similar or different conclusions. As the measurement methods are different, it is totally fine to get different observations. However, it would be essential to discuss relevant previous works for comparison. Three relevant papers are listed below:

"Stochastic variations of migration speed between cells in clonal populations"

"Single-cell migration chip for chemotaxis-based microfluidic selection of heterogeneous cell populations"

"Single-cell RNA-sequencing of migratory breast cancer cells: discovering genes associated with cancer metastasis"

The authors claim that "22.7% of cells tracked (2048 individual cells) met the requirements to be used in the analysis." Excluding majority of the data can be biased. Other reviewers raised similar concerns as well. It is recommended that the authors should better explain how they excluded data? For example, the data were excluded for three possible reasons. x% for A reason, y% for B reason, and z% for C reason. Also, if including those excluded data, will it change the conclusion?

It is recommended that the authors can discuss the possibility that cellular motility was driven by secreted signals (e.g. secreted proteins/exosome). If secreted signals are the dominant drivers of cell motility. Neighboring cells naturally share similar motility. It may not be caused by heritability as suggested by the authors. Further data analysis maybe necessary to eliminate this concern.

Review form: Reviewer 2

Is the manuscript scientifically sound in its present form?

Yes

Are the interpretations and conclusions justified by the results?

Yes

Is the language acceptable?

Yes

Do you have any ethical concerns with this paper?

No

Have you any concerns about statistical analyses in this paper?

No

Recommendation?

Accept as is

Comments to the Author(s)

Congratulations on a very nice piece of work. I look forward to citing it.

Review form: Reviewer 3**Is the manuscript scientifically sound in its present form?**

Yes

Are the interpretations and conclusions justified by the results?

Yes

Is the language acceptable?

Yes

Do you have any ethical concerns with this paper?

No

Have you any concerns about statistical analyses in this paper?

Yes

Recommendation?

Accept with minor revision (please list in comments)

Comments to the Author(s)

I am satisfied with the authors' replies, bar two minor comments:

1. Could the data showing the speed of cells that move off-screen before before dividing (0 daughters) vs cells which divide on-screen (min 2 daughters) be shown?

4. Can formula used for the parent-offspring regression be shown? I presume given the reply that h is then the variance - for someone outside the field such as myself this would be useful to see.

Decision letter (RSOS-191645.R0)

21-Jan-2020

Dear Dr Dash,

The editors assigned to your paper ("Cancer cell lines show high heritability for motility but not generation time") have now received comments from reviewers. We would like you to revise your paper in accordance with the referee and Associate Editor suggestions which can be found below (not including confidential reports to the Editor). Please note this decision does not guarantee eventual acceptance.

Please submit a copy of your revised paper before 13-Feb-2020. Please note that the revision deadline will expire at 00.00am on this date. If we do not hear from you within this time then it will be assumed that the paper has been withdrawn. In exceptional circumstances, extensions

may be possible if agreed with the Editorial Office in advance. We do not allow multiple rounds of revision so we urge you to make every effort to fully address all of the comments at this stage. If deemed necessary by the Editors, your manuscript will be sent back to one or more of the original reviewers for assessment. If the original reviewers are not available, we may invite new reviewers.

- Data accessibility

If you wish to submit your supporting data or code to Dryad (<http://datadryad.org/>), or modify your current submission to dryad, please use the following link:
<http://datadryad.org/submit?journalID=RSOS&manu=RSOS-191645>

- Competing interests

- Authors' contributions

- Acknowledgements

- Funding statement

Kind regards

on behalf of the Associate Editor, and Professor Catrin Pritchard (Subject Editor)
 openscience@royalsociety.org

Associate Editor's comments:

Thank you for your patience while the paper was reviewed - unfortunately, we had some difficulty in soliciting suitable reviewers to assess your manuscript; however, 3 reviewers have now reported. Two are broadly in favour of publication, but the concerns raised by the first reviewer are substantial and we would ask you to respond to these effectively - the revision you submit will be sent to this reviewer for consultation.

Reviewers' Comments to Author:

Reviewer: 1

Comments to the Author(s)

The manuscript entitled "Cancer cell lines show high heritability for motility but not generation time" by Wass et al. tracked cell migration and proliferation of hundreds of cell family from multiple cells to test whether motility and proliferation rate are inheritable. The authors did a lot of work and provided interesting data. However, there are limitations in this study. The authors didn't compare what they found with relevant previous works. In addition, there are concerns regarding data processing/selection and alternative interpretation of the collected data. It is recommended that this paper can be revised before being accepted. The detailed comments are listed below:

1. There were some previous works regarding whether the cell motility can be inherited in the literature. The work from Yan et al is a short-term study more similar to the presented work. They saw low correlation between two daughter cells. The works from Chen et al are long-term heritability studies. They showed cellular motility and altered gene expression can be inherited. It

is recommended that the authors should explicitly discuss what's new in the present work and also why the authors got similar or different conclusions. As the measurement methods are different, it is totally fine to get different observations. However, it would be essential to discuss relevant previous works for comparison. Three relevant papers are listed below:

"Stochastic variations of migration speed between cells in clonal populations"

"Single-cell migration chip for chemotaxis-based microfluidic selection of heterogeneous cell populations"

"Single-cell RNA-sequencing of migratory breast cancer cells: discovering genes associated with cancer metastasis"

The authors claim that "22.7% of cells tracked (2048 individual cells) met the requirements to be used in the analysis." Excluding majority of the data can be biased. Other reviewers raised similar concerns as well. It is recommended that the authors should better explain how they excluded data? For example, the data were excluded for three possible reasons. x% for A reason, y% for B reason, and z% for C reason. Also, if including those excluded data, will it change the conclusion?

It is recommended that the authors can discuss the possibility that cellular motility was driven by secreted signals (e.g. secreted proteins/exosome). If secreted signals are the dominant drivers of cell motility. Neighboring cells naturally share similar motility. It may not be caused by heritability as suggested by the authors. Further data analysis maybe necessary to eliminate this concern.

Reviewer: 2

Comments to the Author(s)

Congratulations on a very nice piece of work. I look forward to citing it.

Reviewer: 3

Comments to the Author(s)

I am satisfied with the authors' replies, bar two minor comments:

1. Could the data showing the speed of cells that move off-screen before before dividing (0 daughters) vs cells which divide on-screen (min 2 daughters) be shown?

4. Can formula used for the parent-offspring regression be shown? I presume given the reply that h is then the variance – for someone outside the field such as myself this would be useful to see.

Author's Response to Decision Letter for (RSOS-191645.R0)

See Appendix A.

RSOS-191645.R1 (Revision)

Review form: Reviewer 1

Is the manuscript scientifically sound in its present form?

Yes

Are the interpretations and conclusions justified by the results?

Yes

Is the language acceptable?

Yes

Do you have any ethical concerns with this paper?

No

Have you any concerns about statistical analyses in this paper?

No

Recommendation?

Accept as is

Comments to the Author(s)

The authors addressed the comments raised by the reviewers.

Review form: Reviewer 3

Is the manuscript scientifically sound in its present form?

Yes

Are the interpretations and conclusions justified by the results?

Yes

Is the language acceptable?

Yes

Do you have any ethical concerns with this paper?

No

Have you any concerns about statistical analyses in this paper?

No

Recommendation?

Accept as is

Comments to the Author(s)

I'm satisfied with the author's revised manuscript. Congrats on a nice study.

Decision letter (RSOS-191645.R1)

06-Mar-2020

Dear Dr Dash,

It is a pleasure to accept your manuscript entitled "Cancer cell lines show high heritability for motility but not generation time" in its current form for publication in Royal Society Open Science. The comments of the reviewer(s) who reviewed your manuscript are included at the foot of this letter.

Kind regards,

Anita Kristiansen
Editorial Coordinator

on behalf of Catrin Pritchard (Subject Editor)
openscience@royalsociety.org

Reviewer comments to Author:
Reviewer: 1

Comments to the Author(s)
The authors addressed the comments raised by the reviewers.

Reviewer: 3

Comments to the Author(s)
I'm satisfied with the author's revised manuscript. Congrats on a nice study.

Appendix A

Response to Reviewers

We thank the reviewers and Editor for their time and their helpful suggestions.

Please find our point-by-point responses to review, and details of amendments to the manuscript, below. Editor's and reviewers' comments are in bold type.

Reviewer 1

1.1: There were some previous works regarding whether the cell motility can be inherited in the literature. The work from Yan et al is a short-term study more similar to the presented work. They saw low correlation between two daughter cells. The works from Chen et al are long-term heritability studies. They showed cellular motility and altered gene expression can be inherited. It is recommended that the authors should explicitly discuss what's new in the present work and also why the authors got similar or different conclusions. As the measurement methods are different, it is totally fine to get different observations. However, it would be essential to discuss relevant previous works for comparison. Three relevant papers are listed below:

"Stochastic variations of migration speed between cells in clonal populations"

"Single-cell migration chip for chemotaxis-based microfluidic selection of heterogeneous cell populations"

"Single-cell RNA-sequencing of migratory breast cancer cells: discovering genes associated with cancer metastasis"

We now cite all three of these studies in the Introduction (page 3, lines 34-38). In the Discussion, we also briefly consider differences in methods and analysis that could explain the discrepancy between our results and those reported by Yan and Irima, who in 2014 tracked sister cell pairs and reported that their speeds were uncorrelated (page 10, lines 206-214).

The reviewer is correct that any number of differences in the experimental method could be responsible for the discrepancy. We now mention a few of these, along with a point about the difference in our datasets and analyses that might also have contributed. I reproduce Figure 4d from Yan and Irima (left). They report that they found no significant correlation between the speed of sister cells D1 and D2, but the trendline on the graph (present in the original Figure) suggests that a correlation may have

been present, although it fell below their chosen threshold of significance. We include slightly more data points in all our regression analyses, allowing for greater sensitivity.

The experimental and statistical explanations are not mutually exclusive, and both could be in play.

The method of Yan and Irima would have allowed for the calculation of heritability from sister cell pairs, and the microfluidic techniques described in the Chen papers could have been used in an experiment to calculate heritability in cell populations (by statistical analysis of the phenotypes of a large sample of cell lines each founded from a single progenitor), but this parameter was not estimated by either group, as it was not the focus of the papers.

1.2: The authors claim that "22.7% of cells tracked (2048 individual cells) met the requirements to be used in the analysis." Excluding majority of the data can be biased. Other reviewers raised similar concerns as well. It is recommended that the authors should better explain how they excluded data? For example, the data were excluded for three possible reasons. x% for A reason, y% for B reason, and z% for C reason. Also, if including those excluded data, will it change the conclusion?

We have reworded the MS where appropriate (page 6, lines 104-107) to more clearly explain that our only reason for not including cells in the analysis is that they moved off-screen before sufficient cell generations could be tracked. We therefore analysed all cell families in which it was possible to track at least one pair of "cousin cells", and so we hold no excluded data which could sensibly be re-included in any new analysis that estimated the extent of stably inherited variation in phenotype. We now explain this rationale more clearly and carefully. However, we have still included the 22.7% figure for the sake of transparency.

We have further addressed the underlying issue highlighted by this comment (and also mentioned by reviewer 3; see point 3.1 below) by acknowledging more explicitly (page 7, lines 119-126) that it remains a possibility that faster cells are slightly under-represented in our dataset. We have also included the results of an additional statistical test, comparing the speed of individual cells that divided vs. those that moved offscreen. There was no significant difference, indicating that while we cannot rule out this possibility altogether, its effects will not be large. We have tried to keep the discussion of this point concise, but we hope that we have now explained this limitation sufficiently clearly for a broad readership.

1.3: It is recommended that the authors can discuss the possibility that cellular motility was driven by secreted signals (e.g. secreted proteins/exosome). If secreted signals are the dominant drivers of cell motility. Neighboring cells naturally share similar motility. It may not be caused by heritability as suggested by the authors. Further data analysis maybe necessary to eliminate this concern.

We agree that secreted signals could be another possible reason for the higher similarity between very closely related cells which have yet to spatially diverge, and we now briefly discuss this possibility in the Introduction (page 5, lines 72-75) and in the Results & Discussion (line 165). However, when comparing cousin cells, larger spatial scales are involved (see cell speeds and division rates from Table 1), and given that cells are covered in liquid growth medium in which the diffusion of secreted signals will be rapid, cousin cells are unlikely to be within signalling range of one another. Secreted signals may indeed

contribute to overall environmental variation affecting motility phenotype, as cells could be moving in and out of range of one another - this is an intriguing possibility, but one that we feel is outside the scope of this particular paper.

Reviewer 2

Congratulations on a very nice piece of work. I look forward to citing it.

We thank Reviewer 2.

Reviewer 3

I am satisfied with the authors' replies, bar two minor comments:

3.1: Could the data showing the speed of cells that move off-screen before before dividing (0 daughters) vs cells which divide on-screen (min 2 daughters) be shown?

In response to this query and the related query from reviewer 1 (point 1.2 above) we went back to our raw data and tested, over all cells tracked, whether cell speed was significantly different for cells that moved offscreen before division compared to cells that did not. There was no significant difference, and we now report this result in the text of the manuscript (lines 119-126). For the convenience of the reviewers, we also present this extra data below. Error bars show two standard deviations from the mean. Note that the units of speed here are calculated using pixels so are not the same units as in Table 1, and also that no MCF7 cells moved offscreen.

	Divided mean speed	Divided stdev	Moved off screen mean speed	Moved off screen stdev
HT1080:	37.55	10.032	40.97	10.87
MDA	26.83	11.37	31.38	11.04
MCF7	10.76	3.74	NA	NA
HeLa	12.48	5.23	12.29	4.98

3.2: Can formula used for the parent-offspring regression be shown? I presume given the reply that h is then the variance – for someone outside the field such as myself this would be useful to see.

The heritability is the slope of a simple linear regression, which we now clarify, and we now include the formula for this quantity, expressed in terms of the variance and covariance of cell speeds (lines 129-133). We feel this is more intuitive for most readers than expressing the same quantity in terms of individual x and y values. However, both formulae can be found in statistics textbooks, and the quantitative genetics textbook we cite in reference to the technique (Lynch & Walsh, reference 28, cited on line 135) also includes extensive discussion of applications, limitations and modifications of parent-offspring regression methods for heritability estimation.